# Patient Safety Incidents in Primary Care: Comparing APEAS–2007 (Spanish Patient Safety Adverse Events Study in Primary Care) with Data from a Health Area in Catalonia (Spain) in 2019

**DOI:** 10.3390/healthcare12111086

**Published:** 2024-05-25

**Authors:** Montserrat Gens-Barberà, Maria-Pilar Astier-Peña, Núria Hernández-Vidal, Immaculada Hospital-Guardiola, Ferran Bejarano-Romero, Eva Mª Oya-Girona, Yolanda Mengíbar-Garcia, Nuria Mansergas-Collado, Angel Vila-Rovira, Sara Martínez-Torres, Cristina Rey-Reñones, Francisco Martín-Luján

**Affiliations:** 1Quality and Patient Safety Central Functional Unit, Gerència d’Atenció Primària Camp de Tarragona, Institut Català de la Salut, 43005 Tarragona, Spainfbejarano.tgn.ics@gencat.cat (F.B.-R.); nmansergas.tgn.ics@gencat.cat (N.M.-C.);; 2QiSP-Tar Research Group, Fundació Institut Universitari per a la Recerca a l’Atenció Primària de Salut—IDIAP Jordi Gol, 08007 Barcelona, Spainfmartin.tgn.ics@gencat.cat (F.M.-L.); 3ISAC Research Group (Intervencions Sanitàries i Activitats Comunitàries; 2021 SGR 00884), Fundació Institut Universitari per a la Recerca a l’Atenció Primària de Salut—IDIAPJGol, 08007 Barcelona, Spain; 4Universitas Health Center, Health Service of Aragon, 50080 Zaragoza, Spain; 5Research Support Unit Camp of Tarragona, Department of Primary Care Camp de Tarragona, Institut Català de la Salut, 43202 Reus, Spain; 6Department of Medicine and Surgery, School of Medicine and Health Sciences, Universitat Rovira i Virgili, 43201 Reus, Spain

**Keywords:** patient safety, primary healthcare, voluntary patient safety event reporting, risk management, healthcare quality

## Abstract

The initial APEAS study, conducted in June 2007, examined adverse events (AEs) in Spanish Primary Healthcare (PHC). Since then, significant changes have occurred in healthcare systems. To evaluate these changes, a study was conducted in the Camp de Tarragona PHC region (CTPHC) in June 2019. This cross-sectional study aimed to identify AEs in 20 PHC centres in Camp de Tarragona. Data collection used an online questionnaire adapted from APEAS–2007, and a comparative statistical analysis between APEAS–2007 and CTPHC–2019 was performed. The results revealed an increase in nursing notifications and a decrease in notifications from family doctors. Furthermore, fewer AEs were reported overall, particularly in medication-related incidents and healthcare-associated infections, with an increase noted in no-harm incidents. However, AEs related to worsened clinical outcomes, communication issues, care management, and administrative errors increased. Concerning severity, there was a decrease in severe AEs, coupled with an increase in moderate AEs. Despite family doctors perceiving a reduction in medication-related incidents, the overall preventability of AEs remained unchanged. In conclusion, the reporting patterns, nature, and causal factors of AEs in Spanish PHC have evolved over time. While there has been a decrease in medication-related incidents and severe AEs, challenges persist in communication, care management, and clinical outcomes. Although professionals reported reduced severity, the perception of preventability remains an area that requires attention.

## 1. Introduction

Patient safety (PS) is widely recognized as a fundamental component of healthcare quality and is a global priority in health policies [1]. The Global Action Plan on PS defines PS as a comprehensive framework of activities aimed at creating cultures, processes, behaviours, technologies, and environments that consistently and sustainably reduce risks, minimize avoidable harm, prevent errors, and mitigate the impact of harm when it occurs [2]. This framework underscores the significance of implementing risk management strategies within healthcare organizations to identify and address risks across various healthcare contexts.

Clinical risk management encompasses two perspectives: proactive and reactive risk management. In terms of reactive risk management, PS reporting and learning systems (PSRLSs) play a pivotal role in addressing and deriving lessons from incidents [3]. The Global Patient Safety Action Plan 2021–2030 highlights the importance of implementing PSRLSs as a valuable tool for healthcare organizations to improve PS, learn from errors, and foster the development of safer practises, ultimately enhancing the quality and safety of healthcare [2].

Studies conducted in hospital settings suggest that incidents occur in approximately 9–12% of admissions, with an associated preventability rate of 80% [4]. These incidents commonly involve the diagnostic process, medication administration, technical procedures, and surgeries. In contrast, PS incidents in Primary Healthcare (PHC) settings are generally less frequent. An international systematic review indicates that 2 or 3 PS incidents are identified for every 100 PHC visits, with 1 in 25 incidents causing serious harm to the patient [5]. The most impactful errors in PHC are related to diagnoses and medication [5]. Several publications provide information about the most frequent errors in PS in PHC [6,7,8,9].

In Spain, the first national study on the impact of adverse events (AEs) in PHC was the APEAS study, conducted in June 2007 [10]. This prospective observational study involved 48 PHC centres from 16 autonomous communities over a two-week period. At that time, Spain lacked PSRLSs, so the data were collected through handwritten self-declaration reports. The APEAS–2007 study, which included 96,047 patients, showed a prevalence of 1.12% AEs in all the PHC visits during the study period. The study emphasized the importance of AE prevention in PHC. Despite seemingly low incidence rates, the number of affected patients was still high, with even a medical doctor who makes 20 consultations a day potentially encountering a harmful incident within a week. Additionally, up to 70% of AEs could have been prevented. Severity and preventability were positively correlated, indicating that severe incidents were preventable with the appropriate measures [11].

In 2013, the Spanish Ministry of Health initiated the implementation of the National PSRLSs in PHC across 11 out of 18 regional health systems. Additionally, seven regional health systems developed their own PSRLSs, including Andalusia, the Balearic Islands, the Basque Country, Castile and Leon, Catalonia, Madrid, and Valencia [12]. These systems provide updated information on PS incidents in the National Healthcare System. However, the utilization of PSRLSs in healthcare organizations is closely tied to the PS culture within those entities [13,14]. A national study on PS culture among PHC professionals in Spain unveiled a positive global index of PS culture, though dimensions related to rhythm and workload were the lowest [15]. Thus, an increase in the rhythm and workload within PHCs is likely to influence the PS culture of professionals.

The reporting of PS incidents in PHC remains relatively low [16]. Simultaneously, since the APEAS–2007 study, other contributing factors to PS, such as increased technology and telemedicine usage, along with a growing population of complex patients, draw a different risk map in PHC [17]. Therefore, a new prospective study could provide insights into the evolving epidemiology of PS incidents in PHC.

In accordance with this, PHC centres in the Camp de Tarragona Health Region of the Catalan Institute of Health allocate specific resources to report and analyze PS incidents through a dedicated reporting and learning system [18]. Over time, this region has demonstrated a high PS culture with a high rate of reported incidents. In 2018, the Spanish national notification system, which includes the participation of 153 PHC centres, reported 737 incidents [19]. In contrast, the Camp de Tarragona region, encompassing 26 PHC centres, reported 1315 incidents, showcasing its high notification capacity. Additionally, the percentage of AEs reported was comparable between the national data (19.15%) and regional data (18.99%) [20]. For these reasons, notifications of incidents and AEs reported in the Camp de Tarragona region could be comparable to those reported in the APEAS–2007 study, providing new data about the notification of AEs.

The current study aims to analyze and describe the types of AEs reported and their contributing factors in the PHC centres of Camp de Tarragona through a cross-sectional approach, and to compare the results with the APEAS–2007 study. This analysis seeks to inform the development of a new risk management plan for PHC.

## 2. Materials and Methods

### 2.1. Study Design and Setting

A prospective descriptive study of all incidents reported during the two central weeks of June 2019 (ten working days) in the Camp de Tarragona PHC centres of the Catalan Institute of Health (CTPHC–2019 study) was conducted. This study was realized in the framework of the APEAS–2007 study within the research project mAPaSP (Study of a Patient Safety Risk Map in Primary Health Care).

PHC in the Camp de Tarragona Health Region of the Catalan Health Institute serves 54.1% of the region’s population (328,945 inhabitants). PHC is provided through 20 PHC Teams, with 73 local clinics, 1 Paediatric Healthcare Team, 2 Sexual and Reproductive Healthcare Teams, 2 PHC Emergency Centres, and 1 Healthcare Team in the regional prison.

### 2.2. Incident Reporting and Data Collection Procedure Project

A procedure using mAPaSP was built to standardize the participation of all PHC professionals in Camp de Tarragona. Briefly, it consisted of two steps:

#### 2.2.1. Step (1): Incident Detection through mAPaSP Study Collaborators

In every contact with a patient, professionals reported whether there was an incident related to PS during the defined study period of two weeks in June 2019. If an incident related to PS was detected, it was registered in the online mAPaSP application or on a paper form. Afterwards, the same professionals or a designated person recorded the paper forms into the online mAPaSP application.

The incident-reporting system in each centre was overseen by a healthcare professional who monitored the number of incidents reported and encouraged staff to stay vigilant. This professional was part of the unit’s team and was responsible for implementing the quality and patient safety strategy.

#### 2.2.2. Step (2) Registration of mAPaSP Study Incidents in the Catalan Patient Safety Incident Reporting System Cloud Platform

Incidents reported in the mAPaSP application were registered in the regional reporting system through a cloud platform accessible on the corporate Intranet of the Tarragona Regional Management of the Catalan Health Institute [19,20]. Notably, this information was automatically stored (including a backup copy).

The questionnaire used to collect data via the online platform of the mAPaSP study in CTPHC–2019 was structured in several steps and was like that of the APEAS–2007 study. Further details about the questionnaire can be found elsewhere [20] and in Appendix A. The questionnaire included all the variables of the APEAS–2007 study with the WHO Patient Safety Taxonomy recommendations of 2009. Additional items were included, encompassing patient risk factors, descriptions of chronic complex patients, end-of-life considerations, and the Adjusted Morbidity Groups. Regarding location, some options were added, such as community healthcare, laboratory, nursing homes, specifying the type of visit (face-to-face, at home, or telehealth), and front-desk services. Considering causal factors, vaccines were included in the category of medication errors.

Professionals were encouraged to record any situation that could indicate a PS incident. These reports were confidential, non-punitive, and entered on a voluntary basis. Each case received an identification code that allowed its follow-up. Once the incident was closed, the information was stored anonymously. Access to the management system was password-protected.

### 2.3. Taxonomy and Definition of Variables

Briefly, all incidents were categorized simultaneously according to the classification proposed by the World Health Organisation (WHO) and the Health Department Accreditation Model for PHC [21,22,23]. The severity of the incident was categorized based on the WHO definitions proposed for PHC. Causal factors were classified according to the proposal of the APEAS–2007 study, separating them into five groups related to communication problems, management, care, diagnosis, and medication [10,11]. Contributing factors were classified into five categories related to the professional, the patient, the organization, the work environment, and external factors. Finally, the preventability of the PS incidents was assessed at the discretion of the notifying professional, categorized as potentially avoidable, unavoidable, or doubtful.

In Appendix A, the variables in the notification registry of PS incidents are detailed.

To ensure comparability, only the 20 Primary Healthcare (PHC) Teams were included in the CTPHC 2019 study, excluding specific population services such as women, children under 15 years old, and individuals in prison.

### 2.4. Statistical Analysis

A descriptive analysis of categorical and quantitative variables was conducted. Categorical variables were described by their frequency distribution and continuous variables were described by the mean and standard deviation (SD) or median, first quartile and third quartile, depending on whether or not they had a normal distribution, respectively. A comparison of CTPHC–2019 and APEAS–2007 studies was performed regarding only the adverse events. To detect differences between the studies, χ2 or Fisher’s test were used for categorical variables and Student’s t-test or ANOVA were used for quantitative variables.

All analyses were carried out using the statistical package R (R Foundation for Statistical Computing, Vienna, Austria, 2018), version 3.4.4. The level of significance was set at *p* < 0.05.

## 3. Results

Throughout the study period, APEAS–2007 included 96,047 visits and 450 PHC professionals, whereas CTPHC–2019 included 24,560 and 179, respectively. A comparison of participant categories between both studies revealed a significant increase in nurse participation in CTPHC–2019 (*p* < 0.001), alongside a decrease in the participation of family doctors and paediatricians. Similar trends were observed when comparing PHC visits between both studies. An increase in nurse visits was observed in CTPHC–2019, whereas a decrease in family doctors’ and paediatricians’ visits was reported (Figure 1).

### 3.1. Adverse Events: Prevalence and Nature

Overall, CTPHC–2019 reported fewer incidents than APEAS–2007 (2.23% vs. 0.60%; *p* < 0.001), particularly in AEs (51.66% vs. 7.13%; *p* < 0.001), while an increase in no-harm incidents was revealed (34.92% vs. 79.97%; *p* < 0.001). Upon comparing the reporting of AEs between both studies, CTPHC–2019 showed a general decrease in AEs compared to APEAS–2007 (1.12% vs. 0.43%; *p* < 0.001), especially in those AEs reported by family doctors (1.04% vs. 0.59%; *p* < 0.001) and nurses (1.15% vs. 0.22%; *p* < 0.001). In terms of the nature of AEs, a change in the frequency of AEs related to medication and those associated with healthcare infections was observed between CTPHC–2019 and APEAS–2007. However, an increase in AEs related to a worsened clinical course of the underlying disease and those associated with healthcare-related infections was observed in CTPHC–2019, being the most prevalent type of AE (Table 1).

### 3.2. Adverse Events: Causal Factors

In the comparison of causal factors associated with AE notifications, both studies identified medication as the most prevalent causal factor. Although the absolute numbers of AEs were different between both studies, their distribution depending on the causal factors was similar. However, in the CTPHC–2019 study, there were higher numbers of AEs related to healthcare (25.72% vs. 32.38%; *p* < 0.001), communication (24.64% vs. 36.19%; *p* < 0.001), diagnosis (13.09% vs. 24.76%; *p* < 0.001), and management (8.94% vs. 29.52%; *p* < 0.001) compared to the APEAS–2007 study (Figure 2).

### 3.3. Severity and Preventability

When comparing both periods, significant overall differences were observed in the severity distribution of AEs. In CTPHC–2019, any mild AEs were reported, whereas a significant increase in moderate AEs and a slight decrease in severe AEs were reported in comparison to APEAS–2007 (Table 2).

Taking into account the severity of the AEs reported by professionals, in the APEAS–2007 study, most AEs were categorized as mild, while in the CTPHC–2019 study they were mostly considered moderate, regardless of the professional category. Specifically, in APEAS–2007, AEs reported by family doctors were classified as 57.04% mild, 36.87% moderate, and 6.1% severe, while those reported by nurses were classified as 50.82%, 38.52% and 10.66%, respectively. Otherwise, in the CTPHC–2019 study, AEs reported by family doctors were classified as 97.26% moderate and 2.74% severe, while those reported by nurses were all classified as moderate.

Regarding the preventability of AEs and their nature, no differences were observed between both studies in the global distribution. However, preventability classification based on professional categories showed statistically significant differences in those AEs related to medication among family doctors (Table 3).

Globally, administrative staff and paediatricians were taken into account, but a comparison was established between family doctors and nurses, who were described in the APEAS–2007 study. The differences between the APEAS–2007 and CTPHC–2019 studies were assessed by the Chi-square or Fisher’s tests.

Analyzing the probability of prevention for the AEs notified, an increase in AEs with no probability, high probability, and total probability of prevention was observed in CTPHC–2019 compared to APEAS–2007 (*p* < 0.001). However, a decrease in AEs with low and moderate probability of prevention was reported (*p* < 0.001) (Table 4).

## 4. Discussion

A cross-sectional study of AE notifications in the Camp de Tarragona Primary Healthcare (PHC) centres was conducted in June 2019 (the CTPHC–2019 study) using a methodology similar to the APEAS–2007 study. This study aimed to compare the results from both studies to identify changes over time in the types of AEs and their contributing factors in PHC. Overall, the findings indicated that AEs did not increase over time. In the CTPHC–2019 study, there was a noticeable increase in AE reporting by nurses, and for the first time, administrative staff were involved in AE reporting. A significant shift in the nature of AEs was observed, with an increase in procedure-related AEs and a decrease in medication-related AEs. Regarding severity, moderate AEs increased in CTPHC–2019, and the global perception of AE preventability decreased.

The PS culture in Catalonia received a significant boost in 2012 with the establishment of the Catalan model for PS in PHC, which aimed to support risk management within PHC teams [24]. Currently, all PHC professionals are involved in PS culture, including administrative staff, who were not included in APEAS–2007. The engagement of administrative staff allows for dealing with patients’ identification issues in different phases of the care process. It also facilitates the management of patients’ demands, bureaucratic procedures, referrals, and appointments to ensure an effective and efficient patient care process [25]. However, when compared to APEAS–2007, the inclusion of incidents reported by administrative staff introduces a confounding factor, as it changes the profile of notifications. The varying levels of experience and knowledge among different professional categories directly influence the types of incidents they identify and report [26]. The reduced proportion of notifications related to serious AEs and issues associated with diagnosis and medication may be attributed to a lower participation of family physicians and paediatricians in reporting compared to the APEAS study.

In addition to the inclusion of administrative staff, in the CTPHC–2019 study, there was an increased participation of nurses compared to APEAS–2007. This increased involvement of nurses, along with the incorporation of administrative staff, may reflect the development of advanced competencies and heightened awareness of the importance of risk management in their daily tasks within PHC teams. Notably, nursing advanced competencies encompass chronic patient management, acute demand management, prevention and promotion programmes, and community involvement. These competences have been successfully developed in Catalonia [27]. Thus, it could be argued that there is a paradigm shift in PHC to deal with this new context. For this reason, it is necessary to design and implement strategies that foster a positive PS culture to avoid punitive responses and to effectively apply and evaluate these changes [28].

Different perceptions among healthcare professionals could be a confounding factor. The variation in reporting rates between nurses and physicians was attributed to differing definitions of what constitutes a PS incident or AE [29]. Previous results in the region also showed that physicians tended to report adverse events more frequently than other healthcare professionals [18]. Additionally, regarding the culture of incident reporting, it is worth noting that higher reporting rates do not necessarily indicate poorer patient care; rather, they reflect a proactive organizational approach to encouraging incident reporting [30].

Our results demonstrated a shift in the PS incident types reported by PHC professionals. There was an increased number of PS incidents without harm reported, coupled with a decrease in the AEs reported by both family doctors and nurses. These results could be attributed to the risk management strategy implemented in the territory, contributing to an enhanced PS culture among PHC professionals in the CTPHC–2019 study, as mentioned before. According to this, as previously mentioned, each PHC team designates a quality and PS referent responsible for driving initiatives in PS and healthcare risk management [31]. On the one side, PHC quality and safety referents promote proactive risk management activities through a technological tool with clinical environment checklists to ensure safe conditions for care provision [32]. On the other side, they reactively handle PS incident notifications through a cloud platform, analysing and redesigning processes and procedures, and implementing safe practises in a team learning process [19]. This organizational structure for risk management in PHC teams not only benefits patients and their families, but also the professionals themselves and the organization at large. Moreover, this organizational structure promotes a stronger patient safety culture, which could be related to increased incident reporting and vice versa [33,34,35,36].

This new risk map allows us to redirect PS management and provides an opportunity to implement targeted measures and practises to reduce risk and improve patient care [18].

AEs related to medication continue to represent the most critical area. Furthermore, AEs related to communication and care management have seen a notable increase. Communication is one of the primary contributors to adverse events in clinical practise [37]. In addition, the attitudes and behaviours of healthcare workers play a key role in establishing a proper safety culture [38]. It has been found that incidents are three times more frequently related to the organization of healthcare than to the knowledge and skills of healthcare professionals, particularly concerning the workflow in general practitioners’ offices and communication between providers and with patients [39]. In that sense, a systematic review highlights the significance of using communication tools to improve patient safety in the centres [40].

Furthermore, in the comparative analysis, the perception of the preventability of AEs has remained constant, despite the identification of fewer AEs. This underscores the necessity for enhanced research to provide guidance to PHC professionals in addressing this issue. For instance, recent studies on medication safety incidents linked to remote PHC delivery have uncovered common incident types associated with electronic prescriptions [41]. This emphasizes the significance of investigating and improving practises to ensure PS in the evolving landscape of healthcare delivery.

The rise of precision medicine regarding genetic or biological characteristics of patients, coupled with advancements in artificial intelligence, can improve clinical risk management [42]. While we acknowledge that AEs are not completely preventable, their cumulative incidence can certainly be reduced through learning-based policies, PHC practise interventions, and including patients and families in a safe healthcare journey [43,44,45].

Therefore, it is crucial for PHC organizations to update clinical risk maps through reporting and learning systems to be aware of new risks that harm patients. PS in PHC consists effectively managing risks to maximize benefit and minimize harm over the course of a patient’s life and disease progression [2].

## 5. Conclusions

The prevalence, nature, and causal factors of AEs in PHC have evolved over the years. Although medication-related issues and severe AEs are not as frequently reported as in 2007, there is a notable increase in challenges related to communication, care management, and disease course exacerbation. While professionals reported fewer severe AEs, the unchanged perception of preventability underscores a persistent concern that requires ongoing attention and intervention.

## Figures and Tables

**Figure 1 healthcare-12-01086-f001:**
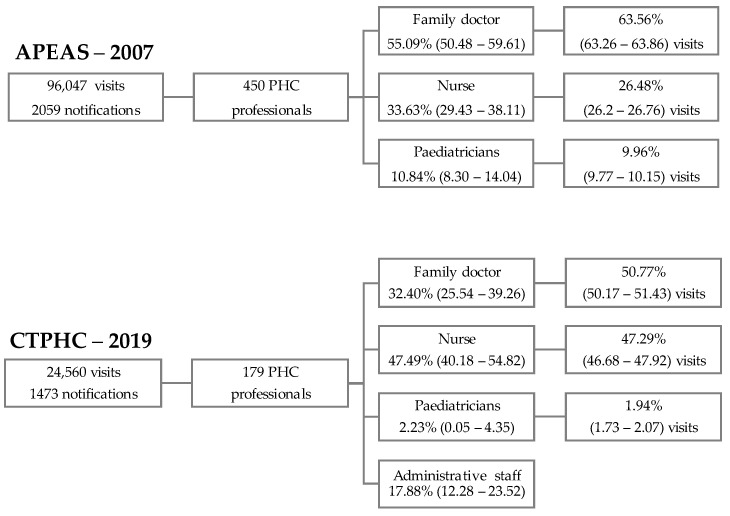
Flowchart of the number of visits and notifications and the distribution of PHC professionals’ participation in the APEAS–2007 and CTPHC–2019 studies. PHC; Primary Healthcare.

**Figure 2 healthcare-12-01086-f002:**
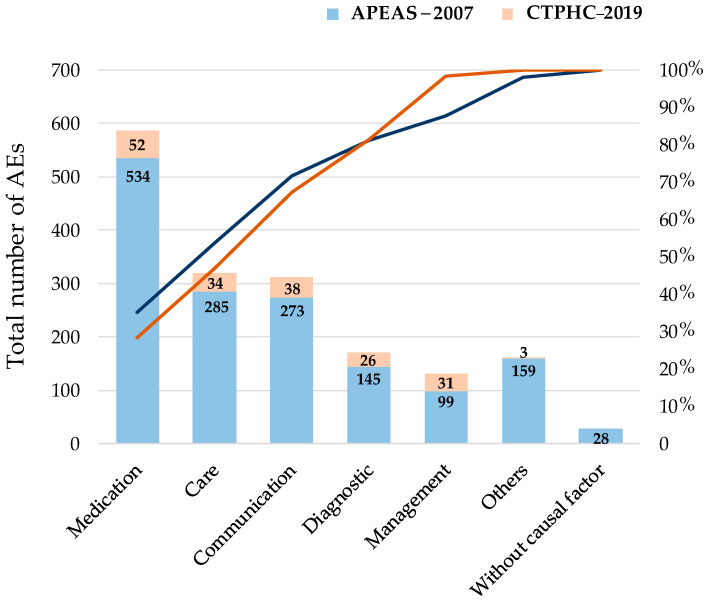
Pareto chart of the total and frequency of adverse events categorized according to the causal factors in the APEAS–2007 and CTPHC–2019 studies. AEs: adverse events.

**Table 1 healthcare-12-01086-t001:** Comparative analysis between the APEAS–2007 study and CTPHC–2019 study regarding the nature of adverse events.

Nature of AEs	APEAS–2007		CTPHC–2019		*p*-Value
	N = 1108	% (IC 95%)	N = 165	% (IC 95%)	
Related to medication	530	47.83 (44.91–50.78)	46	27.88 (21.04–34.72)	<0.001
Worsened clinical course of the underlying disease	221	19.95 (17.70–22.40)	65	39.39 (31.94–46.85)	
Related to procedures	118	10.65 (8.97–12.60)	23	13.94 (8.65–19.22)	
Associated with healthcare-related infections	93	8.39 (6.90–10.17)	2	1.21 (0.03–2.88)	
Related to care	72	6.50 (5.19–8.11)	9	5.45 (1.99–8.92)	
Others	74	6.68 (5.35–8.30)	20	12.12 (7.14–17.10)	

Difference between APEAS–2007 and CTPHC–2019 studies assessed by Chi-square test.

**Table 2 healthcare-12-01086-t002:** Reported adverse events according to their severity and PCP in the APEAS–2007 and CTPHC–2019 studies.

Grade of Severity	APEAS–2007		CTPHC–2019		*p*-Value
	N = 1108	% (IC 95%)	N = 105	% (IC 95%)	
Mild	606	54.69 (51.76–57.62)	-	-	*p* < 0.001
Moderate	421	38.00 (35.14–40.85)	103	98.1 (95.48–99.48)	
Severe	81	7.31 (5.78–8.84)	2	1.90 (0.52–4.52)	

Difference between APEAS–2007 and CTPHC–2019 studies assessed by Chi-square test.

**Table 3 healthcare-12-01086-t003:** Comparative analysis of preventability according to the professional category and the nature of the adverse events in the APEAS–2007 and CTPHC–2019 studies.

	APEAS–2007		CTPHC–2019		*p*-Value
	N	% Avoidable Events	N	% Avoidable Events	
Total PCP					
Related to procedure	94	79.7	20	86.96	0.567
Associated with healthcare-related infections	74	79.6	2	100.0	1.0
Related to care	52	72.2	8	88.89	0.434
Related to medication	313	59.1	17	43.59	0.085
Worsened clinical course of the underlying disease	183	82.8	49	90.74	0.219
Other factors	79	83.8	17	85.0	1.0
Total	778	70.2	113	76.87	0.116
Family doctors					
Procedure	33	75.0	14	87.5	0.481
Infection	26	74.3	2	100.0	1.0
Healthcare	12	70.6	3	100.0	0.540
Medication	229	58.0	13	40.63	0.003
Worsened evolutionary course of the disease	131	81.4	31	88.57	0.439
Other factors	31	83.8	11	84.62	1.0
Total	462	67.1	74	73.27	0.308
Nurse					
Procedure	60	82.2	4	80.0	1.0
Infection	47	82.5	-	-	
Healthcare	39	73.6	4	80.0	1.0
Medication	70	64.2	4	66.67	1.0
Worsened evolutionary course of the disease	37	86.0	15	93.75	0.661
Other factors	27	87.1	5	83.33	1.0
Total	280	76.5	22	84.21	0.382

**Table 4 healthcare-12-01086-t004:** Comparative analysis of the probability of prevention of adverse events in the APEAS–2007 and CTPHC–2019 studies.

Nature of AE	APEAS–2007		CTPHC–2019		*p*-Value
	N = 1108	% (IC 95%)	N = 165	% (IC 95%)	
Related to medication	530	47.83 (44.91–50.78)	46	27.88 (21.04–34.72)	<0.001
Worsened clinical course of the underlying disease	221	19.95 (17.70–22.40)	65	39.39 (31.94–46.85)	
Related to procedures	118	10.65 (8.97–12.60)	23	13.94 (8.65–19.22)	
Associated with healthcare-related infections	93	8.39 (6.90–10.17)	2	1.21 (0.03–2.88)	
Related to care	72	6.50 (5.19–8.11)	9	5.45 (1.99–8.92)	
Others	74	6.68 (5.35–8.30)	20	12.12 (7.14–17.10)	

Difference between APEAS–2007 and CTPHC–2019 studies assessed by Chi-square test.

## Data Availability

Data are contained within the article.

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
