# Peer review of "Patient Safety Incidents in Primary Care: Comparing APEAS–2007 (Spanish Patient Safety Adverse Events Study in Primary Care) with Data from a Health Area in Catalonia (Spain) in 2019"

_healthcare, 2024, doi:10.3390/healthcare12111086_

Round 1

Reviewer 1 Report

Comments and Suggestions for Authors

The paper is a study about patient safety and adverse events. The methods are defined and the design criteria seems correct. However, further statistical analysis and visualization of findings, and some case-studies/outliers will further improv the quality of this work.

Reviewer 2 Report

Comments and Suggestions for Authors

Dear Authors,

CTPHC-2019 was conducted in primary healthcare settings based on a previous APEAS-2007 study. The authors used nearly identical parameters and compared the results, such as adverse events, generalizability, and severity. This cross-sectional study is essential to developing a comprehensive risk management strategy. The study is commendable, but there are some significant comments that the authors need to address in the manuscript.

1. Figure 1 displays two flow charts: (1) The upper chart depicts CTPHC-2019, and (2) the lower chart illustrates APEAS-2007. However, the captions' order is reversed despite the evident difference in visit counts. Could you reorder them or annotate the flowchart accordingly?

2. The CTPHC-2019 focus on a single region within Spain may limit the generalizability of its findings to other areas. PHC systems, cultural contexts, and reporting practices can vary significantly, which may affect the transferability of the results. It limits the generalizability of the present study. How do the authors justify this point?

3. The CTPHC-2019 categorized incidents based on severity, causal factors, and preventability. The discussion lacks an exploration of the confounding factors influencing incident reporting trends. Understanding these factors could provide deeper insights into the root causes of PS incidents.

4. The authors didn’t explore the reasons behind changes in incident patterns, which could provide valuable insights. Do the authors offer any interpretations regarding this matter?

5. The APEAS-2007 and CTPHC-2019 provide a quick picture of PS incidents during a specific period, which may not capture the seasonal variations but need a longitudinal analysis to track trends and changes over time. Longitudinal data could reveal patterns, identify emerging risks, and assess the effectiveness of interventions more effectively. Long-term follow-up studies are needed to determine the effectiveness of interventions in improving PS outcomes in PHC. Do the authors have any plans to extend the study in that direction?

The authors should thoroughly explain each point to cover all aspects in detail.

Thanks

Reviewer 3 Report

Comments and Suggestions for Authors

Thank you for the opportunity to review "Patient Safety Incidents in a Primary Care: comparing APEAS-2007 (Spanish patient safety adverse events study in primary 3 care) with data from a health area in Catalonia (Spain) in 2019" submitted to Healthcare. This work is important, but the manuscript requires a major revision as the content is dense and difficult to review due to what I believe is a lack of clarity in presentation and the need for more concise statements. There are multiple incongruencies in statements from the introduction to the results. However, I recognize the difficulty in reporting this type of study. For this reason, I highly recommend the authors review the STROBE checklist for cross-sectional studies (https://www.equator-network.org/reporting-guidelines/strobe/) to remember the reporting requirements for a cross-sectional study with an analytical instrument design.

In the next section, I am providing notes for the authors to consider for areas to focus when revising the manuscript. In addition, I have attached a file with comments and feedback. Although the comments may appear critical, they are fully intended to be constructive in letting the authors know where I was struggling to understand the current manuscript in reference to the APEAS-2007 (although I thought the study was published in 2008) which I have worked with secondary data from this study. My expectation is the authors will be able to use a majority of the feedback, but not necessarily all of the recommendations, to make a successful revision.

--------------------------

NOTES:

--------------------------

The introduction is too dense with information that is not relevant to the current study. For this reason, the authors are advised to provide a proper two to three paragraph introduction to clearly and concisely present the rationale and significance for the current study. Then, the other information can be moved to a background section. There is very little information about the systematic reviews specific to the topic of patient safety in primary care cited in the introduction. Why is the a comparison of the APEAS-2007 and the current data necessary and important? Although the manuscript states, "This analysis seeks to inform the development of a new risk management plan for PHC" yet the discussion did not seem to focus on this information. Where is the current study analysis informing the development of a new risk management plan?

--------------------------

There needs to be increased clarity in the similarities and differences between the CTPHC-2019 study in comparison to the APEAS-2007. After reviewing the two studies, and the Spanish literature cited in the manuscript, there appear to be substantive differences beginning with the questionnaire, possibly with the sample, and seemingly in the analysis. Also, the APEAS-2007 study is not specifically cited in the current study as proxy studies published by the same authors are cited. For example, please consider the incongruent statements from the abstract to the results in terms of the purpose and methods of the study.

ABSTRACT: "This cross-sectional study aimed to identify AEs in 20 PHC centres in Camp de Tarragona. Data collection used an online questionnaire adapted from APEAS-2007, and a comparative statistical analysis between APEAS-2007 and CTPHC-2019 was performed."

CLOSE OF INTRODUCTION: "The current study aims to replicate the APEAS-2007 cross-sectional study within the PHC centres of Camp de Tarragona. The objective is to analyse the shifts in the types of AEs reported and the contributing factors compared to APEAS-2007."

METHODS: "This study replicated the APEAS-2007 study methods [15]."

ANALYSIS: "Comparison of CTPHC-2019 and APEAS-2007 studies was performed regarding only the adverse events."

Then, the results provide many additional comparisons despite there being multiple differences in the CTPHC-2019 study in comparison to the APEAS-2007. Again, please clarify what was done, why it was done, and how it was done.

CONCLUSION: "Although medication-related issues and severe AEs are not as frequently reported as in 2007, there is a notable increase in challenges related to communication, care management, and disease course exacerbation."

This is not clearly linked to the purpose, analysis, and findings in terms of the comparative.

--------------------------

The discussion section seems to be disjointed from the results section and drift away from the original stated purpose of the study. Also, I recommend adding literature from the global context of patient safety in primary healthcare as this is an international journal.  

--------------------------

The similarities and differences in the methods for the current study with the APEAS-2007 need to be clearly presented. This is especially important specific to the revisions to the questionnaire for the current study.

--------------------------

Six of the 31 references, or 20%, are auto-citations. This is a reason I noted the lack of global references external to Spain in the discussion section. Please make sure to incorporate the relevant global literature into the manuscript in the introduction and discussion sections. 

--------------------------

Please reconcile the author contribution listing with the authors. The first author, for example, is not listed in the author contributions. In addition, the contributions are not reported according to the journal requirements.

--------------------------

The trademark statements for The Patient Safety Company™ and TPSCloud™ are not necessary in this manuscript as these are not explained to have any special function differing from other cloud based data collection systems. Simply stating "a cloud platform" is the standard in health information technology for reporting this information. 

--------------------------

As the QiSP-Tar Research Group has research funded by pharmaceutical companies and there are several brands and trademarks (e.g. The Patient Safety Company™ and TPSCloud™) noted in multiple publications by the Group, I want to ask the authors to make sure there are no conflicts of interest as none are stated.

Comments on the Quality of English Language

The language could benefit from a solid editing focused on the form, flow, and style to polish the presentation.

Round 2

Reviewer 1 Report

Comments and Suggestions for Authors

The authors responded well to my comments and I recommend publication of this manuscript.

Reviewer 2 Report

Comments and Suggestions for Authors

Thanks for addressing all the comments.